# The Antifungal Effect of *Weissella confusa* WIKIM51 (Wilac D001) on Vaginal Epithelial Cells Infected by *Candida albicans*

Gain Lee [1,2], Young-Ah You [2], Abuzar Ansari [2], Yoon-Young Go [2], Sunwha Park [2], Young Min Hur [2], Soo-Min Kim [1], Sang Min Park [3] and Young Ju Kim [1,2,*]

[1] Graduate Program in System Health Science and Engineering, Ewha Womans University, Seoul 03760, Republic of Korea; loveleee0102@gmail.com (G.L.); zeus_0218@ewhain.net (S.-M.K.)

[2] Department of Obstetrics and Gynecology, Ewha Medical Research Institute, College of Medicine, Ewha Womans University, Seoul 07984, Republic of Korea; yayou@ewha.ac.kr (Y.-A.Y.); abu.kim.0313@gmail.com (A.A.); gokogoko@ewha.ac.kr (Y.-Y.G.); clarrissa15@gmail.com (S.P.); hym1210@ewha.ac.kr (Y.M.H.)

[3] Pharmsville Co., Ltd., Seoul 06642, Republic of Korea; plan1@pharmsville.com

[*] Correspondence: kkyj@ewha.ac.kr

**Abstract:** Vulvovaginal candidiasis (VVC) is a genital infection caused by *Candida albicans* (*C. albicans*). *Weissella confusa* WIKIM51 (Wilac D001) is known to be detected in dandelion kimchi, produce lactic acid, and have an anti−inflammatory ability; however, its diverse antifungal effects have not been studied. Here, we investigated the antifungal effect of Wilac D001 in *C. albicans* compared to *Lactobacillus* species on vaginal epithelial cells (VECs). To test the antifungal ability of Wilac D001 against *C. albicans* on VECs, an adhesion test, pro-inflammatory cytokines (interleukin (IL)-6 and IL-8) analysis, and a disk diffusion test were performed. The acid tolerance test was conducted to investigate the viability of Wilac D001 in various acidic conditions. *Lactobacillus reuteri (L. reuteri)* and *L. rhamnosus* were used as positive controls. Wilac D001 showed the capacity to inhibit the colonization of *C. albicans* by adhering to VECs, with an inhibitory effect similar to that of positive controls. Both pro−inflammatory cytokines including IL−6 and IL−8 concentrations were significantly decreased when Wilac D001 was treated on *C. albicans*-infected VECs, respectively ($p < 0.001$). The result of the disk diffusion test indicates that the inhibitory ability of Wilac D001 is comparable to *L. reuteri* and *L. rhamnosus* on agar plates infected with *C. albicans*. Our results demonstrate that *Weissella confusa* WIKIM51 has antifungal effects against VECs infected by *C. albicans*.

**Keywords:** vulvovaginal candidiasis; *Candida albicans*; vaginal epithelial cells; *Weissella confusa* WIKIM51; *Lactobacillus reuteri*; *Lactobacilus rhamnosus*

## 1. Introduction

Vulvovaginal candidiasis (VVC), caused by *Candida* species, is the second most common fungal infection of the genital tract [1]. *Candida albicans* (*C. albicans*), the major etiologic agent of VVC, is present, with a prevalence of up to 78% [2,3]. The onset of VVC is mostly associated with the use of antibiotics, lack of *Lactobacillus* spp. in the vagina, sexual activity, and use of hygiene products [2,4]. The common symptoms of VVC include itching, burning, and redness [5].

Antibiotic therapy is an effective clinical strategy for controlling *C. albicans* infection. However, the continuous use of antibiotics can increase drug resistance, relapse rates, and interrupt the composition of vaginal microbiota [6,7]. Recent in vitro and in vivo studies have demonstrated lactobacilli-mediated immune defense against *C. albicans* colonization [8,9]. Thus, interest in the development of alternative strategies that involve indirect treatment of VVC using lactobacilli has increased.

*Lactobacillus* species represent the predominant bacterial group in healthy vaginas. *Lactobacillus* species are a major source of lactic acid and bacteriocins that maintain the vaginal

microflora in an acidic environment, contributing to reducing the invasion of pathogenic infections by competing to the adhesion site [10,11]. Moreover, these vaginal lactobacilli regulate the innate immune system through controlling the secretion of cytokines and antibody responses against pathogenic infections [12]. Prior in vivo and in vitro studies on *Lactobacillus* spp. have reported that *L. reuteri* and *L. rhamnosus* can modulate inflammatory immune responses to counteract against *C. albicans* [13–15].

The *Weissella* was separated from *Lactobacillus* and categorized as a genus in 1993 [16]. *Weissella confusa* WIKIM51 (Wilac D001) was isolated from dandelion kimchi, a traditional Korean fermented food [17]. Several studies have revealed that *Weissella confusa* produces diverse components such as lactic acid, exopolysaccharide (EPS), and bacteriocin, which facilitate anti-inflammatory and antioxidant activities in the colon [18–20]. Moreover, *Weissella*-abundant women deliver at full term in *bacteriodes*-dominated communities [21]. The oral administration of *Weissella* into mice showed that the abundances of *Lactobaillus* species were increased and modulated regulatory T cells [22,23]. However, the antifungal effect of Wilac D001 has not been fully explored in comparison with *Lactobacillus* spp. in the vagina. In this study, we examined the antifungal efficacy of Wilac D001 against *C. albicans* infection in vaginal epithelial cells (VECs) and compared it with that of *L. reuteri* and *L. rhamnosus*.

## 2. Materials and Methods

### 2.1. Fungal and Bacterial Strains

The pathogenic yeast *C. albicans* was purchased from the Korean Collection of Type Cultures (Seoul, Republic of Korea; KCTC7270). The *C. albicans* was cultured using yeast malt (YM) broth medium (Becton, Dickinson and Company, Franklin Lakes, NJ, USA) under the aerobic condition for 24 h at 30 °C [24].

Bacterial strains, including *Weissella confusa* WIKIM51 (Wilac D001), *L. reuteri* (RC−14), and *L. rhamnosus* (GR−1), were supplied by Pharms Ville (Seoul, Republic of Korea). In this study, *L. rhamnosus* and *L. reuteri* were considered positive controls. De Man, Rogosa, and Sharpe (MRS) broth (Becton, Dickinson and Company, Franklin Lakes, NJ, USA) were used to culture bacteria, including Wilac D001 and the positive controls. The bacterial strains were cultured under the conditions described in Supplementary Table S1.

### 2.2. Culture of Vaginal Epithelial Cell

The vaginal epithelial cell line (VK2/E6E7, CRL−2616) was purchased from the American Type Culture Collection (ATCC; Manassas, VA, USA). The VECs were cultured in collagen (Sigma, St. Louis, MO, USA)-coated 75 $cm^2$ cell culture flasks using keratinocyte-serum-free medium (K-SFM; GIBCO-BRL San Giuliano Milanese, Milan, Italy) supplemented with components provided by the manufacturer. Finally, the media supplemented with 0.4 mM $CaCl_2$ (Sigma, St. Louis, MO, USA) and 500 UI/mL penicillin (Sigma, St. Louis, MO, USA) were used. The VECs were maintained at 37 °C in a 5% $CO_2$ incubator. The culture medium was replaced every 2−3 days, and cell viability was assessed using trypan blue dye (Sigma, MO, USA) and a hemocytometer (Paul Marienfeld GmbH & Co. KG, Lauda-Königshofen, Gemany).

### 2.3. Adhesion Test

Adhesion tests were conducted following a previously standardized method [25]. For this assay, a sterilized cover glass was placed in each well of a 6−well plate to facilitate the observation of morphological changes in VECs using a light microscope (BX41; Olympus, Japan). Wilac D001 and *C. albicans* ($1 \times 10^7$ CFU/mL) were added to VECs, followed by their incubation for 1 h. Subsequently, a Gram staining (Sigma) assay was performed, and the frequency of adherent cells was calculated [26]. The number of adherent VECs was determined for 30 consecutive cells in each replication [27]. The percentage of adherent cells (%) was calculated using the following formula:

$$Adherent\ Cell\ (\%) = \frac{VECs\ attached\ bacteria\ without\ C.\ albicans}{Total\ cells} \times 100$$

### 2.4. Enzyme-Linked Immunosorbent Assay (ELISA) Analysis

VECs were seeded at a density of $1 \times 10^5$ cells/mL in 6-well plates and maintained at 37 °C in a 5% $CO_2$ environment for 96 h. To test whether Wilac D001 can regulate pro−inflammation cytokine, VECs were incubated with *C. albicans* ($1 \times 10^7$ CFU/100 μL) for 12 h and sequentially replaced by Wilac D001 ($1 \times 10^7$ CFU/100 μL) for 24 h. VECs were co−cultured with *C. albicans* ($1 \times 10^7$ CFU/100 μL) and maintained for 36 h to induce inflammation [28]. Next, the sample in each well was washed with phosphate-buffered saline (PBS; Biosesang, Gyeonggi-do, Republic of Korea), and $1 \times 10^5$ CFU/mL of Wilac D001 was added to it. After co-culturing with Wilac D001 for 24 h, the supernatant of the medium was carefully collected and centrifuged at 5000 rpm to remove cell debris. The expression of IL−6 and IL−8 in the supernatant was quantified using ELISA kits (Abbkine, Atlanta, Georgia, USA) according to the instructions provided by the manufacturer. The concentrations of IL−6 and IL−8 in each supernatant were measured at 450 nm using a microplate reader.

### 2.5. Disk Diffusion Assay

For the disk diffusion analysis, commonly used to determine the susceptibility of pathogenic microorganisms to antibiotics, the method described by Matevosyan et al. [29] was followed with modifications. *C. albicans* was cultured using YM broth for 18 h at 30 °C in a shaking condition (100 rpm). Subsequently, *C. albicans* cultures were washed twice with PBS. A suspension of *C. albicans* ($1 \times 10^5$ CFU) was evenly spread on the MRS agar plate and incubated for 3 h at 30 °C. Next, sterile paper disks (diameter: 5 mm) were placed onto the agar plate, and a bacterial suspension of $1 \times 10^7$ CFU/25 μL was added to each paper disk, followed by incubation for 24 h at 37 °C. A zone of inhibition (ZoI) was produced around the disk area in which *C. albicans* did not grow. *Sertaconazole nitrate,* as antimicrobial agent, was used as a positive control [29].

### 2.6. Statistical Analysis

Statistical analyses were performed using the SPSS statistical package (version 20.0; SPSS Inc., Armonk, NY, USA). One-way analysis of variance (ANOVA) and Student's *t*-test were used to compare the results between different experimental groups and positive controls. All experiments were independently repeated 2 or 3 times. The Shapiro–Wilk test was performed to test normal distribution, and all data were satisfied normality. The statistical significance was considered at a *p*-value < 0.05.

## 3. Results

### 3.1. Protective Effect of Wilac D001 against C. albicans on VECs

We observed the monolayer of VECs after 96 h of incubation (Figure 1A). The detachment of VECs was observed following colonization by *C. albicans* (Figure 1B). To investigate whether Wilac D001 competitively protects VECs against *C. albicans,* Wilac D001 and *C. albicans* were co−cultured for 1 h on VECs (Figure 2A). No significant differences were observed between Wilac D001 and the positive controls (Figure 2B). The VECs were surrounded by Wilac D001, and they showed any colonization by *C. albicans* (Figure 2E and Figure S1). Similar outcomes were observed during the co−incubation of VECs with *C. albicans* and the positive control strains *L. reuteri* and *L. rhamnosus* (Figure 2C,D).

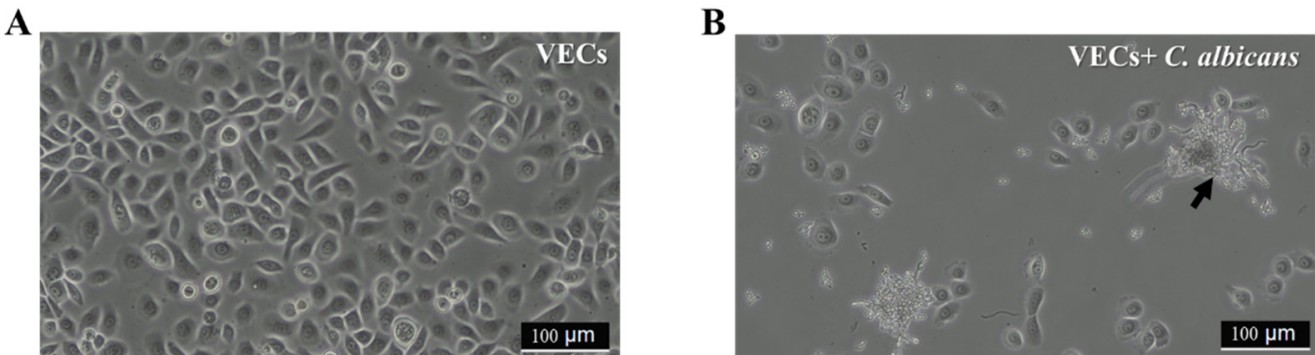

**Figure 1.** Morphological images of vaginal epithelial cells (VECs) and VECs infected by *Candida albicans (C. albicans)*. (**A**) The microscopic images of monolayer VECs after VECs were 90–100% confluent for 96 h and (**B**) infectious VECs by $1 \times 10^7$ CFU of *C. albicans*. Black arrow indicates the aggregation of *C. albicans*. VECs = vaginal epithelial cells, *C. albicans* = *Candida albicans*, scale bar = 100 μm.

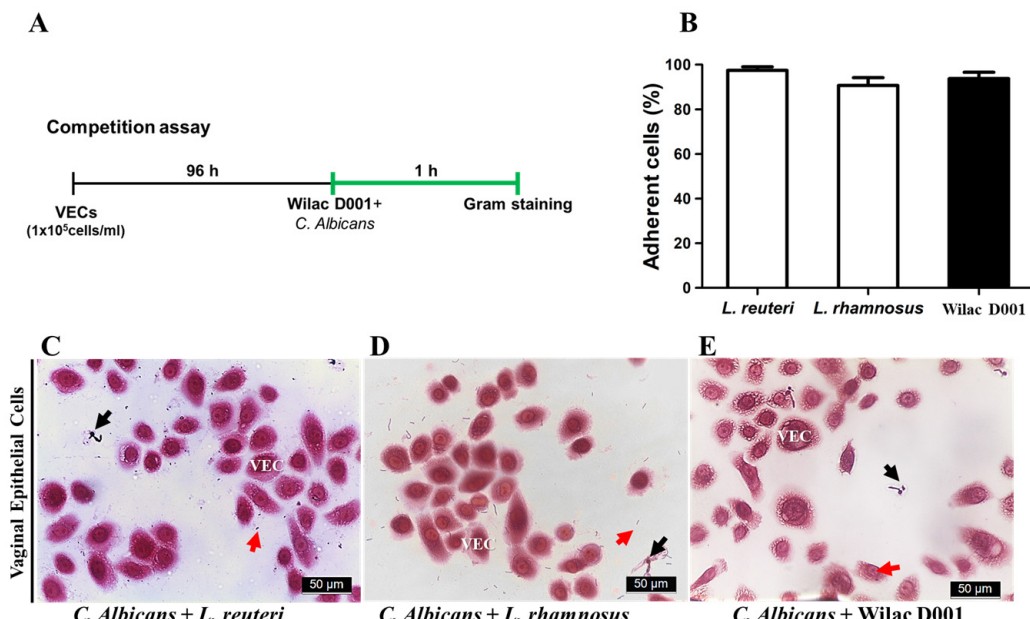

**Figure 2.** Wilac D001 prevented the attachment of *C. albicans* on VECs. (**A**) Competition assay indicated that Wilac D001 and *C. albicans* were inoculated for 1 h after VECs were 90–100% confluent for 96 h. (**B**) The percentage of adherent cells was determined in *L. reuteri*, *L. rhamnosus*, and Wilac-D001-treated VECs. The vaginal epithelial cells with (**C**) *L. reuteri*, (**D**) *L. rhamnosus*, and (**E**) Wilac D001 were cultured with *C. albicans*, respectively, and then Gram staining was performed. The black arrow indicates *C. albicans*, and red arrow indicates respective bacteria strains. Data were analyzed using a one-way analysis of variance (ANOVA). Bar graphs are expressed as mean ± SD. Scale bar = 50 μm. VECs: vaginal epithelial cells.

### 3.2. The Inhibitory Effect of Pro−Inflammatory Cytokine Production

VECs infected by *C. albicans* were cultured for 24 h in the presence or absence of Wilac D001. Subsequently, the concentrations of IL−6 and IL−8 were assessed using the ELISA method (Figure 3A). The upregulated IL−6 and IL−8 expression levels in VECs exposed to *C. albicans* were suppressed up to approximately two-fold after treatment with Wilac D001 (Figure 3B,C). Additionally, *L. reuteri* reduced IL−6 and IL−8 production in VECs infected with *C. albicans* (Figure 3D,E). However, the presence of *L. rhamnosus* did not significantly affect the production of these cytokines (Figure 3F,G).

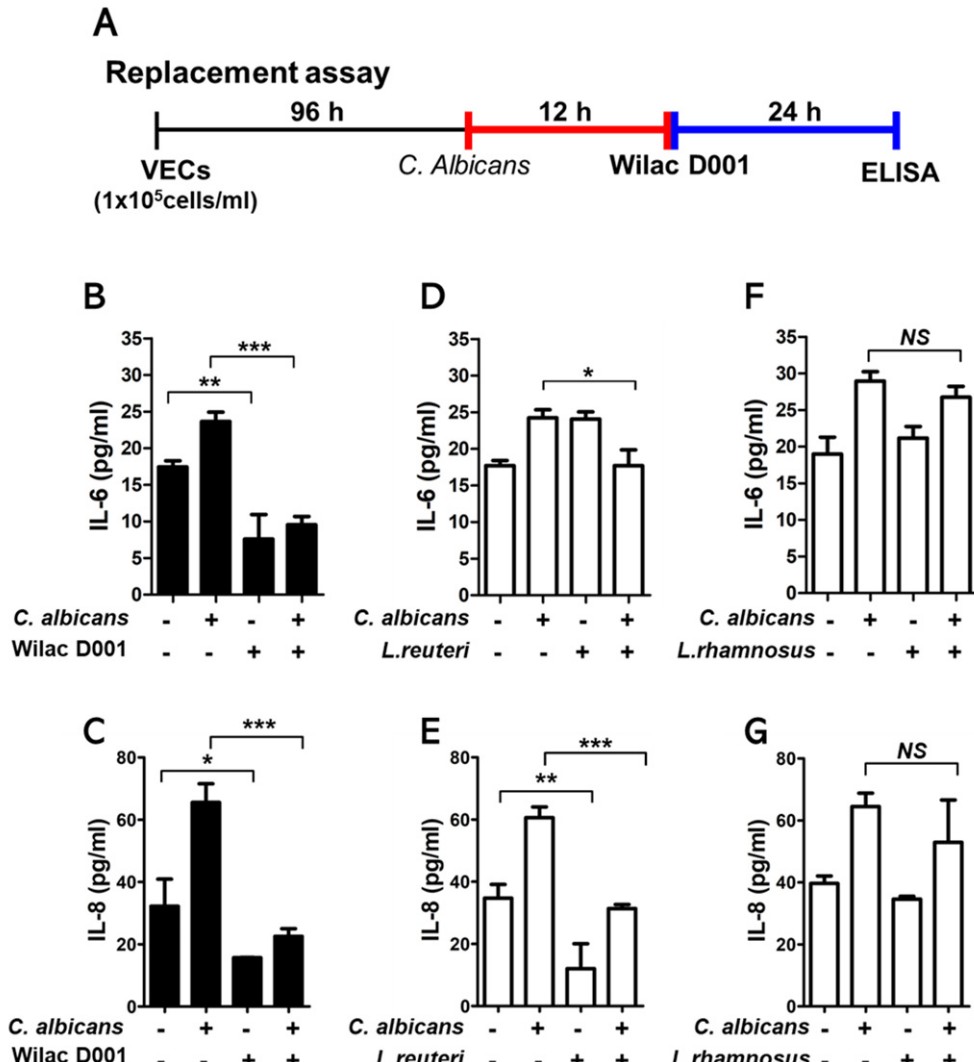

**Figure 3.** Wilac D001 reduced the expression levels of pro−inflammatory cytokines on infectious VECs with *C. albicans*. (**A**) Replacement assay showed that *C. albicans* and Wilac D001 were incubated serially on confluent VECs for ELISA assay. VECs were cultured with *C. albicans* in the presence or absence of Wilac D001, and the concentration of IL−6 and IL−8 was measured. Representative graphs show concentrations of IL−6 (**B,D,F**) and IL−8 (**C,E,G**) produced by VECs with Wilac D001 (**B,C**), *L. reuteri* (**D,E**), and *L. rhamnosus* (**F,G**). The results are described as Mean ± SD from triplicate experiments. VECs = vaginal epithelial cells, ELISA = enzyme-linked immunosorbent assay. Data were analyzed using a one-way analysis of variance (ANOVA) with Bonferroni's post hoc correlation. *NS*: not significant. *: *p*-value < 0.05, **: *p*-value < 0.01, ***: *p*-value < 0.001. *n* = 3.

### 3.3. Susceptibility of Wilac D001 against C. alibcans

Next, to identify the susceptibility of *C. albicans* to Wilac D001, the ZoI was measured to estimate the ability of Wilac D001 to inhibit the growth of *C. albicans* [27]. The antimicrobial agent (*Sertaconazole nitrate*) and *Lactobacillus* spp. were used as positive controls in this assay. All bacteria, including Wilac D001, demonstrated a significant antifungal effect compared to the blank disk. The ZoI exhibited by Wilac D001 confirmed its inhibitory effect, which is comparable to that of *L. reuteri* and *L. rhamnosus* (Table 1). However, the inhibitory effect of the antibiotic was greater than that of all bacterial strains.

**Table 1.** Diameters of the ZoI formed by Wilac D001 and all tested positive controls.

| | Negative | Positive Control | | | Wilac D001 |
|---|---|---|---|---|---|
| | *Blank* | *Sertaconazole Nitrate* | *L. rueteri* | *L. rhamnosus* | |
| ZoI (diameter, mm) | 6.0 ± 0.0 | 18.3 ± 0.6 | 10.8 ± 0.3 *,# | 11.2 ± 0.3 *,# | 11.2 ± 0.3 *,# |

The results are indicated as mean ± SD. *Sertaconazole nitrate* was used as the antimicrobial. ZoI: zone of inhibition. Data were analyzed using a one-way analysis of variance (ANOVA) with Bonferroni's post hoc correlation. *: $p < 0.001$ compare with blank, #: $p < 0.001$ compare with *Sertaconazole Nitrate*, $n = 3$.

## 4. Discussion

Our results indicate that *Weissella confusa* WIKIM51 (Wilac D001) exhibits antifungal activity by inhibiting the formation of *C. albicans* biofilm. Pro−inflammatory cytokines, including IL−6 and IL−8, were upregulated by *C. albicans* infection in VECs; however, they were significantly downregulated by Wilac D001. The inhibitory effect of Wilac D001 against the growth of *C. albicans* was similar to that of *L. reuteri* and *L. rhamnosus.* These results propose that the antifungal effects of Wilac D001 against *C. albicans* are comparable to those of *L. reuteri* and *L. rhamnosus.* In particular, Wilac D001 exhibits a significantly higher efficiency in regulating pro−inflammatory cytokines than *L. reuteri* and *L. rhamnosus.*

A vaginal environment dominated by *Lactobacillus* spp. is crucial for protecting the genital tract from opportunistic infections [4]. *Lactobacillus* produces lactic acid to maintain an acidic environment and inhibits the growth of other microorganisms as well as the influx of pathogens [4,25]. A lack of lactobacilli in the vagina is associated with infections caused by *Candida* spp., an opportunistic pathogen. Several gynecologic diseases, including pelvic inflammatory disease (PIV), sensitivity to human immunodeficiency virus (HIV), and even adverse pregnancy outcomes, such as the preterm premature rupture of membranes (PPROMs) and preterm birth (PTB), are associated with the proportion of lactobacilli in the vagina [30–32].

Here, we have investigated the antifungal effect of Wilac D001, which was isolated from dandelion kimchi, against *C. albicans* on vascular endothelial cells for the first time. There is concern about the safety of the *Weissella* genus due to potential pathogenicity [33]. Some studies report that certain strains of *W. confusa* may cause infections, particularly in immunocompromised patients [34,35]. However, other *W. confusa* strains show promise in mitigating gut dysbiosis linked to inflammatory bowel disease and cancers, with demonstrated benefits in maintaining intestinal tight junctions and modulating intestinal cell responses [19,35,36]. Therefore, the precise taxonomic category is crucial for understanding its potential pathogenicity. The strain that we used in this experiment is Wilac D001, which is a non−spore−forming, Gram−positive coccobacillus that has obtained safety approval from the FDA. Wilac D001 has been annotated by the National Center for Biotechnology Information (NCBI, CP110106).

The biofilm formation is the essential mechanism underlying *C. albicans* infection; it enhances pathogenicity through increased colonization and microbial adherence [37]. Adhesion to the surface of the epithelial cell is pivotal to the constant colonization of *C. albicans* [26,38]. Hyphae−morphological colonization was observed in vaginal epithelial cells infected with *C. albicans*. However, Wilac D001 protected VECs by blocking the congregation of *C. albicans* on VECs. *Lactobacillus* spp. also reduced the adhesion of *C. albicans* to VECs surfaces by competing for adhesion sites, as reported [39].

The expression of pro−inflammatory cytokines, including IL−6 and IL−8, is upregulated in VECs, owing to *C. albicans* infection [40]. Furthermore, a clinical study demonstrated that the concentration of IL−8 was higher in women infected with *Candida* spp. [41]. Our data indicated that Wilac D001 significantly reduced the production of both IL−6 and IL−8 in VECs infected with *C. albicans*. A previous study reports that the response of inflammatory expressions to lipopolysaccharide stimulation can be reduced in a mild acidic environment (pH 5.5−6.0) [42]. However, we did not analyze culture media pH

produced by culturing probiotics over 24 h. A pH−mediated cytokines investigation might be needed to understand the interaction between pH and cytokine expression.

The diameters of the ZoIs reflect the antifungal efficacies of bacterial strains. Antimicrobial agents diffuse on the agar surface inoculated with *C. albicans* and inhibit its growth [43]. As we found in our results, use of an antimicrobial could be a convincing strategy to reduce the growth of *C. albicans*. However, Wilac D001 and *lactobacillus* spp. present a significant antifungal effect compared to disk only. Considering the disadvantage of the frequent use of antibiotics, such as the increase prevalence and incidence of VVC [44,45], alternative treatment with lactobacilli could be a promising long-term treatment without the development of antibiotic resistance.

There are still some limitations to this study. The antifungal ability of Wilac D001 against VVC was assessed only in vitro. Consequently, an in vivo experiment should be conducted prior to oral administration. Another limitation of this study is that the antifungal efficacy of Wilac D001 was not as pronounced as that of an antimicrobial agent. Nevertheless, the ZoI induced by Wilac D001 was comparable to that formed by *L. reuteri* and *L. rhamnosus*. Lastly, a single microbial was tested independently. Given that human microorganisms comprise numerous bacteria species, an experiment involving a combination with *Lactobacillus* species will be necessary, anticipating a synergistic effect.

## 5. Conclusions

Our findings suggest that Wilac D001 inhibits *C. albicans* growth by disrupting biofilm formation. Additionally, Wilac D001 suppresses the *C. albicans*-mediated upregulation of pro−inflammatory cytokines, including IL−6 and IL−8, in VECs. The antifungal effect of Wilac D001 is comparable to that of *L. reuteri* and *L. rhamnosus*. Therefore, Wilac D001 has an antifungal effect on VECs infected by *C. albicans*, especially by regulating pro−inflammatory cytokines. To investigate its efficacy as a health supplement, further in vivo studies will be needed.

**Supplementary Materials:** The following supporting information can be downloaded at: https://www.mdpi.com/article/10.3390/app14072676/s1, Figure S1: Gram staining of VECs adherent Wilac D001 under microscopy; Table S1: Culture conditions of microorganism.

**Author Contributions:** Conceptualization, G.L. and Y.-A.Y.; Methodology, A.A.; Software, S.P.; Validation, S.M.P. and A.A.; Formal Analysis, S.-M.K.; Investigation, G.L. and Y.-A.Y.; Resources, S.P., Y.M.H. and S.M.P.; Data Curation, Y.M.H. and S.P.; Writing—Original Draft Preparation, G.L.; Writing—Review & Editing, Y.-A.Y., G.L. and Y.-Y.G.; Visualization, A.A.; Supervision, Y.J.K.; Project Administration, Y.J.K.; Funding Acquisition, Y.J.K. All authors have read and agreed to the published version of the manuscript.

**Funding:** This research was funded by the Basic Science Research Program through the National Research Foundation of Korea (NRF-2020R1I1A1A01071955, NRF-2020R1I1A1A01074518), BK21 FOUR (Fostering Outstanding Universities for Research) funded by the Ministry of Education (MOE, Korea), and the National Research Foundation of Korea (NRF-5199990614253, Education Research Center for 4IR-Based Health Care). This research was supported by a grant of the Korea Health Technology R&D Project through the Korea Health Industry Development Institute (KHIDI), funded by the Ministry of Health & Welfare, Republic of Korea (grant number: RS-2023-00266554).

**Institutional Review Board Statement:** Not applicable.

**Informed Consent Statement:** Not applicable.

**Data Availability Statement:** Data is contained within the article or Supplementary Material.

**Conflicts of Interest:** Author Sang min Park was employed by the company Pharmsville Co., Ltd. The remaining authors declare that the research was conducted in the absence of any commercial or financial relationships that could be construed as a potential conflict of interest.

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
