# Peer review of "The Antifungal Effect of Weissella confusa WIKIM51 (Wilac D001) on Vaginal Epithelial Cells Infected by Candida albicans"

_applsci, doi:10.3390/app14072676_

Round 1
Reviewer 1 Report
Comments and Suggestions for Authors
The authors demonstrate that co-incubation of vaginal epithelial cells with C. albicans yeast and the bacteria Wilac D001 leads to reduced release of cytokines IL-6 and IL-8 from the epithelial cells. The authors propose that Wilac D001 could be used to reduce vulvovaginal candidiasis. The study is completely in vitro and lacks a confirmatory step in an animal model. The work presented here has some promise but is still in early stage and must be expanded.
Figure 1. The black arrow is not explained. Also, the morphological images are not well described. What is the significance of showing the lower panel of Figure 1A? Is it to show detachment of VK2 cells or to show adherence of yeast cells to the VK2 cells or both?
The most frustrating aspects of Figure 1 are Panels B and C. Why are they associated with Figure 1, when Panel A was carried out differently than what is described in Panels B and C? I think Panel 1B should be incorporated into Figure 2 and that Panel 1C should be incorporated into Figure 3.
Figure 2: This Figure shows Gram staining of VK2 cells treated with C. albicans and either L. reuteri (Panel A), L. rhamnosus (Panel B), or W. confusa (Panel C). But where is the Gram-stained image of VK2 cells treated with C. albicans alone? Also, the same can be said of Panel D of Figure 2. Where is the bar for adherence for VK2 cells treated with C. albicans alone?
Also, in the Material and Methods, the authors say that this strain is pathogenic, but no further information is provided? Is this strain associated with VVC or with some other kind of candidiasis (such as oral or systemic candidiasis)? Please explain.
The data in Figure 3 is the most balanced in the manuscript and includes all necessary treatment groups.
The statistics in Table 1 are unclear. Do the authors mean that sertraconazole was significantly more active than all the others? The footer only says P<0.05, but compared to what? How is this useful for W. confusa? This limitation should be discussed in the Discussion section. Was there any zone of inhibition for control only? Even if no, the control should be included in the table with a “NA” meaning no zone of inhibition observed.
For the acid tolerance, it would also be advisable to test the acid tolerance of the yeast and to add a curve for C. albicans at each time point. W. confusa was only significantly different from the others at pH 2.0 (2 hr) and pH 7.0 (4 hr and 6 hr). At pH 4.0, there is no statistical symbol showing a difference for W. confusa (perhaps it was omitted by error?) Lastly, the OD readouts for Panels A (OD ~ 0.04) and B (OD ~0.06) are well below that of Panel C (OD ~ 0.3), indicating that these organisms did not grow well in the acidic environment which could limit their usefulness as potential treatments for VVC. This should be discussed as well in the Discussion.
Minor points:
The Abstract has some minor language issues; For example, “against” should be “in”; “by” should be “with”; “decrease” should be “decreased”.
The last sentence of paragraph 2 of the Introduction requires improvement. It lacks clarity.
Figure 2 legend. “Scale bar” is misspelled.
"Acknowledgments: The authors acknowledge Drs. Sang min Park and Pharms ville Co., Ltd. (Republic of Korea) for providing fungi and bacteria strains." This is unusual because Dr. Sang min Park is a co-author of the study, so why the need to acknowledge?
Comments on the Quality of English LanguageThe Abstract has some minor language issues; For example, “against” should be “in”; “by” should be “with”; “decrease” should be “decreased”.
The last sentence of paragraph 2 of the Introduction requires improvement. It lacks clarity.
Figure 2 legend. “Scale bar” is misspelled.
Author Response
Thank you very much for taking the time to review this manuscript-applsci-2844512. Please find the detailed responses below and the corresponding revisions and changes in the re-submitted files.
[Reviewer 1]
The authors demonstrate that co-incubation of vaginal epithelial cells with C. albicans yeast and the bacteria Wilac D001 leads to reduced release of cytokines IL-6 and IL-8 from the epithelial cells. The authors propose that Wilac D001 could be used to reduce vulvovaginal candidiasis. The study is completely in vitro and lacks a confirmatory step in an animal model. The work presented here has some promise but is still in early stage and must be expanded.
→ I appreciated about your valuable suggestion. I agree with your comments that there is a lack of validation steps in animal models. This paper is the in vitro experiments to investigate antifungal activity of WIilac D001 on vaginal epithelial cells. Therefore, I have inserted the limitation at page 9, line 352-354 –
“There are still some limitations. The antifungal ability of Wilac D001 against VVC was assessed only in vitro. Consequently, an in vivo experiment should be conducted prior to oral administration.”
Figure 1. The black arrow is not explained. Also, the morphological images are not well described. What is the significance of showing the lower panel of Figure 1A? Is it to show detachment of VK2 cells or to show adherence of yeast cells to the VK2 cells or both? The most frustrating aspects of Figure 1 are Panels B and C. Why are they associated with Figure 1, when Panel A was carried out differently than what is described in Panels B and C? I think Panel 1B should be incorporated into Figure 2 and that Panel 1C should be incorporated into Figure 3.
→ I have corrected figure 1 legend to explain black arrow - “Black arrow indicates the aggregation of C. albicans”. I also have revised the figure structure and legend under the proper contents in details from Figure 1.
For Figure 1 Panels A and B, I wanted to present the morphological differences between normal layer of VECs and infected VECs by C. albicans, especially the detachment –
“Figure 1. Morphological images of vaginal epithelial cells (VECs) and VECs infected by Candida albicans (C. albicans)”.
As your mention, original panels B and C of Figure 1 moved the figure 2 and 3, respectively.
Figure 2: This Figure shows Gram staining of VK2 cells treated with C. albicans and either L. reuteri (Panel A), L. rhamnosus (Panel B), or W. confusa (Panel C). But where is the Gram-stained image of VK2 cells treated with C. albicans alone? Also, the same can be said of Panel D of Figure 2. Where is the bar for adherence for VK2 cells treated with C. albicans alone?
→ I cultured and stained VECs with C. albicans alone. According to reference [1], We focused on the competitive activity between Wilac D001 and C. albicans on consecutive 30 VECs at that moment. We observed most of cells were detached when C. albicans was treated on VECs as you can see in Figure 1 Panel B.
Reference
- Zárate, G.; Nader‐Macias, M. Influence of probiotic vaginal lactobacilli on in vitro adhesion of urogenital pathogens to vaginal epithelial cells. Letters in Applied Microbiology 2006, 43, 174-180.
Also, in the Material and Methods, the authors say that this strain is pathogenic, but no further information is provided? Is this strain associated with VVC or with some other kind of candidiasis (such as oral or systemic candidiasis)? Please explain.
→ The strain that we used in this experiment is the KCTC7270 originated from oral cavity. C. albicans can asymptomatically colonize on mucosal surface of healthy human such as oral cavity, gut, and vaginal cavity. The overgrowth of C. albicans can lead to systemic candidiasis, oropharyngeal and vulvovaginal candidiasis [2, 3].
Reference
- C.; Schuster, M.G.; Vazquez, J.A.; Walsh, T.J. Clinical practice guideline for the management of candidiasis: 2016 update by the Infectious Diseases Society of America. Clinical Infectious Diseases 2016, 62, e1-e50.
- Nobile, C.J.; Johnson, A.D. Candida albicans biofilms and human disease. Annual review of microbiology 2015, 69, 71-92.
The data in Figure 3 is the most balanced in the manuscript and includes all necessary treatment groups. The statistics in Table 1 are unclear.
Do the authors mean that sertraconazole was significantly more active than all the others? The footer only says P < 0.05, but compared to what?
→ I have corrected the statistical significance as per your suggestion. As you interpreted, the antimicrobial agent exhibited significant antifungal activity according to the ANOVA method. To enhance clarity, I have added a statistical description in the table legend –
“Data were analyzed using a one-way analysis of variance (ANOVA) with Bonferroni post hoc correlation. *: P < 0.001 compare with Blank, #: < 0.001 compare with Sertaconazole Nitrate”
How is this useful for W. confusa?
→ I have described in result parti page 6, line 233-235 –
“All bacteria, including Wilac D001 demonstrated a significant antifungal effect compared to the blank disk.”
Additionally, I have revised the utility of W. confusa in the discussion part page 9, line 328-334 –
“As we can find in this result, antimicrobial can be the convincing strategy to reduce growth of C. albicans. However, Wilac D001 and lactobacillus spp. present significant antifungal effect compared to disk only. Considering the disadvantage of frequent use of antibiotics, such as increase prevalence and incidence of VVC [4, 5], alternative treatment with lactobacilli, can be a promising long-term treatment without the development of antibiotic resistance.”
Reference
- Pirotta, M.V.; Gunn, J.M.; Chondros, P. “Not thrush again!” Women's experience of post‐antibiotic vulvovaginitis. Medical journal of Australia 2003, 179, 43-46.
- Xu, J.; Schwartz, K.; Bartoces, M.; Monsur, J.; Severson, R.K.; Sobel, J.D. Effect of Antibiotics on Vulvovaginal Candidiasis: A MetroNet Study. The Journal of the American Board of Family Medicine 2008, 21, 261-268, doi:10.3122/jabfm.2008.04.070169.
This limitation should be discussed in the Discussion section.
→ Considering your comments, the limitation parts have been inserted in page 9, line 357-361 – “Another limitation of this study is that the antifungal efficacy of Wilac D001 was not as pronounced as that of an antimicrobial agent. Nevertheless, the ZoI induced by Wilac D001 was comparable to those formed by L. reuteri and L. rhamnosus”
Was there any zone of inhibition for control only? Even if no, the control should be included in the table with a “NA” meaning no zone of inhibition observed.
→ I inserted the control blank data as you can see in revised Table1.
|
Negative |
Positive control |
Wilac D001 |
|||
|
Blank |
Sertaconazole Nitrate |
L. |
L. |
||
|
ZoI |
6.0±0.0 |
18.3 ±0.6 |
10.8 ±0.3*, # |
11.2 ±0.3*, # |
11.2±0.3*, # |
The results are indicated as mean ± SD. Sertaconazole nitrate was used as the antimicrobial. ZoI: Zone of inhibition, Data were analyzed using a one-way analysis of variance (ANOVA) with Bonferroni post hoc correlation. *: P < 0.001 compare with Blank, #: < 0.001 compare with Sertaconazole Nitrate, n = 3.
For the acid tolerance, it would also be advisable to test the acid tolerance of the yeast and to add a curve for C. albicans at each time point.
→ Previous study has reported that C. albicans can live in the pH range is 2 to 7 [6]. The purpose of this experiment is whether W. confusa can survive in different acidic conditions to investigate the possibility of survival rate in vagina where the pH is 4.
Reference
- Davis, D.A. How human pathogenic fungi sense and adapt to pH: the link to virulence. Current opinion in microbiology 2009, 12, 365-370.
- confusa was only significantly different from the others at pH 2.0 (2 hr) and pH 7.0 (4 hr and 6 hr). At pH 4.0, there is no statistical symbol showing a difference for W. confusa (perhaps it was omitted by error?)
→ I have corrected figure 4 with extra marks and interpretation “Each value points were tested using paired T-test. *: P < 0.05 verses L. reuteri. †: P < 0.05 versus L. rhamnosus, #: P < 0.05 compare between L reuteri and L. rhamnosus.”
Lastly, the OD readouts for Panels A (OD ~ 0.04) and B (OD ~0.06) are well below that of Panel C (OD ~ 0.3), indicating that these organisms did not grow well in the acidic environment which could limit their usefulness as potential treatments for VVC. This should be discussed as well in the Discussion.
→ While significant differences were observed in Panels A (OD ~ 0.04) and B (OD ~ 0.06), as you pointed out, these distinctions were less discernible within the same range as Panel C (OD ~ 0.3). Consequently, I adjusted the y-axis range to enhance clarity for the reader.
I also have corrected discussion part of acid tolerance test in page 9, line 342-354 –
“At pH2 media, Wilac D001 successively survived for 2 hours. Despite the survival rate was decreased from 4 hour, it still showed similar with L. reuteri. In the pH4, the L. rhamnosus showed significantly high survival rate for 6 hours. Wilac D001 also showed significantly high survival rate at 4 hours. Although it was reduced for 6 h, still Wilac D001 can survive as much as L. reuteri, which indicate that Wilac D001 might survive in the acidic vaginal environment for 6 hours. This indicates that Wilac D001 adapts to Lactobacillus-dominated environment and exerts can exhibit antifungal effects in the infected vagina. Interestingly, in the pH7 condition, might be induced pathogenic infection, Wilac D001 survival rates were significantly higher than L. reuteri and L. rhamnosus at each time point.”
Minor points:
The Abstract has some minor language issues; For example, “against” should be “in”; “by” should be “with”; “decrease” should be “decreased”. The last sentence of paragraph 2 of the Introduction requires improvement. It lacks clarity. Figure 2 legend. “Scale bar” is misspelled. "
→ I have corrected the spells and deleted the acknowledge paragraph.
"Acknowledgments: The authors acknowledge Drs. Sang min Park and Pharms ville Co., Ltd. (Republic of Korea) for providing fungi and bacteria strains." This is unusual because Dr. Sang min Park is a co-author of the study, so why the need to acknowledge?
→ As you comment, the acknowledgment part have been deleted.

Reviewer 2 Report
Comments and Suggestions for Authors
The paper effectively delves into the issue of vulvovaginal candidiasis (VVC) caused by Candida albicans and proposes investigating the antifungal properties of Weissella confusa WIKIM51 (Wilac D001) as a potential treatment. It articulates the rationale behind the study well, emphasizing the pressing need for alternative therapies for VVC. However, I recommend a substantial major revision of the article to address specific areas for improvement. These revisions would greatly enhance the paper's impact and contribution to the field.
1. While the paper measures the inflammatory activity through IL6 and IL8, further elucidation of the underlying mechanisms and potential effects of pH on cytokine secretion would strengthen the study. Including control experiments to assess pH-mediated cytokine secretion would provide valuable context.
2. Discussing the potential pathogenicity of Weissella confusa and its relevance to the study's findings would enrich the discussion. Addressing any potential concerns regarding its pathogenicity and contrasting it with its antifungal properties could provide a more comprehensive understanding.
3. A comparative study analyzing the adhesion mechanisms of Candida albicans with both Weissella confusa and Lactobacillus species would enhance the paper. Exploring potential differences in adhesion mechanisms could offer insights into the effectiveness of Wilac D001 as an antifungal agent.
4. The optical images from ZOI experiment is needed for clear visualization of the data.
5. The growth curve indicates the significant growth inhibition of Wilac D001 with time mostly in acidic conditions so the growth curve with a higher time incubation is needed to show its long-term survival in pH 2 and pH 4.
6. While the study highlights the antifungal effects of Wilac D001 against VECs infected by C. albicans, discussing the broader implications of its efficacy in different clinical scenarios or against other microbial infections would enrich the discussion. Addressing the potential applicability of Wilac D001 beyond VVC could provide a more holistic perspective on its therapeutic potential.
Overall, addressing these points would strengthen the paper and enhance its contribution to the field of antifungal therapy for VVC.
Author Response
Thank you very much for taking the time to review this manuscript-applsci-2844512. Please find the detailed responses below and the corresponding revisions and changes in the re-submitted files.
[Reviewer 2]
- While the paper measures the inflammatory activity through IL6 and IL8, further elucidation of the underlying mechanisms and potential effects of pH on cytokine secretion would strengthen the study. Including control experiments to assess pH-mediated cytokine secretion would provide valuable context.
→ I appreciated about your valuable comment. Unfortunately, we were not considering the pH when analyze the cytokine concentration as followed references [1, 2]. For further experiment, we will measure pH and other cytokine related to VVC also.
And I have corrected the sentences about the elucidation of the underlying mechanisms and potential effects of pH on cytokine secretion in Discussion section on page 9, line 319-325 –
“A previous study reports that inflammatory expressions response to lipopolysaccharide stimulation can be reduced in the mild acidic environment (pH 5.5-6.0) [3]. However, we did not analyze culture media pH produced by culturing probiotics over 24 hours. The pH-mediated cytokines investigation might be needed for understanding interaction between pH and cytokine expression.”
Reference
- Niu, X.-X.; Li, T.; Zhang, X.; Wang, S.-X.; Liu, Z.-H. Lactobacillus crispatus Modulates Vaginal Epithelial Cell Innate Response to Candida albicans. Chinese Medical Journal 2017, 130, 273-279, doi:10.4103/0366-6999.198927.
- SANTOS, Carolina MA, et al. Anti-inflammatory effect of two Lactobacillus strains during infection with Gardnerella vaginalis and Candida albicans in a HeLa cell culture model. Microbiology, 2018, 164, 349-358.
- Hackett, A.; Trinick, R.; Rose, K.; Flanagan, B.; McNamara, P. Weakly acidic pH reduces inflammatory cytokine expression in airway epithelial cells. Respiratory research 2016, 17, 1-9.
- Discussing the potential pathogenicity of Weissella confusa and its relevance to the study's findings would enrich the discussion. Addressing any potential concerns regarding its pathogenicity and contrasting it with its antifungal properties could provide a more comprehensive understanding.
→ We described more about this issue in page 8, line 289-292 in discussion part-
“There is a concern about the safety of strains of W. confusa, due to potential pathogenicity [4]. Some studies report that certain strains of W. confusa may cause infections, particularly in immunocompromised patients [5, 6]”
Reference
- Cheaito, R.A.; Awar, G.; Alkozah, M.; Cheaito, M.A.; El Majzoub, I. Meningitis due to Weissella confusa. The American Journal of Emergency Medicine 2020, 38, 1298. e1291-1298. e1293.
- Cupi, D.; Elvig-Jørgensen, S.G. Safety assessment of Weissella confusa–A direct-fed microbial candidate. Regulatory Toxicology and Pharmacology 2019, 107, 104414.
- Ahmed, S.; Singh, S.; Singh, V.; Roberts, K.D.; Zaidi, A.; Rodriguez-Palacios, A. The Weissella genus: Clinically treatable bacteria with antimicrobial/probiotic effects on inflammation and cancer. Microorganisms 2022, 10, 2427.
- A comparative study analyzing the adhesion mechanisms of Candida albicans with both Weissella confusa and Lactobacillus species would enhance the paper. Exploring potential differences in adhesion mechanisms could offer insights into the effectiveness of Wilac D001 as an antifungal agent.
→ W. confusa including Lactobacillus species have an ability to produce lactic acid, bacteriocin, EPS, and hydrogen peroxide, which exert protect vagina against pathogenic invasion [7,8]. These molecular known to inhibit the colonization of C. albcians hyphal formation to epithelial cells by competing to adhesion site. Especially, lactic acid maintains acidic environment that can decrease the yeast-to-hyphal transition, which is the essential step promotion of VVC progression [9-11].
Reference
- Takano, T.; Kudo, H.; Eguchi, S.; Matsumoto, A.; Oka, K.; Yamasaki, Y.; Takahashi, M.; Koshikawa, T.; Takemura, H.; Yamagishi, Y. Inhibitory effects of vaginal Lactobacilli on Candida albicans growth, hyphal formation, biofilm development, and epithelial cell adhesion. Frontiers in Cellular and Infection Microbiology 2023, 13, 503.
- Amabebe, E.; Anumba, D.O. The vaginal microenvironment: the physiologic role of lactobacilli. Frontiers in medicine 2018, 5, 181.
- Matsubara, V.H.; Wang, Y.; Bandara, H.; Mayer, M.P.A.; Samaranayake, L.P. Probiotic lactobacilli inhibit early stages of Candida albicans biofilm development by reducing their growth, cell adhesion, and filamentation. Applied microbiology and biotechnology 2016, 100, 6415-6426
- Coudeyras, S.; Jugie, G.; Vermerie, M.; Forestier, C. Adhesion of human probiotic Lactobacillus rhamnosus to cervical and vaginal cells and interaction with vaginosis-associated pathogens. Infectious diseases in obstetrics and gynecology 2008, 2008.
- Takano, T.; Kudo, H.; Eguchi, S.; Matsumoto, A.; Oka, K.; Yamasaki, Y.; Takahashi, M.; Koshikawa, T.; Takemura, H.; Yamagishi, Y. Inhibitory effects of vaginal Lactobacilli on Candida albicans growth, hyphal formation, biofilm development, and epithelial cell adhesion. Frontiers in Cellular and Infection Microbiology 2023, 13, 503.
- The optical images from ZOI experiment is needed for clear visualization of the data.
→ Unfortunately, I took a part of the pictures just for preliminary records before analyzing. Here, we can attach brief picture of them according to your comments, however, the quality was not enough to provide in main text.
Figure 1 Antifungal test: Antifungal assay performed by disc diffusion method and zone of inhibition was observed. CFU of C. albicans was (A) 1x105 (left penal) and (B) 1x106 (right penal). O: Negative control (blank), 1: W. confusa, 2: L. reteri, 3: L. rhamnosus, 5: L. rhamnosus (GR1), M: L. reteri+ L. rhamnosus, CD: antimicrobial
- The growth curve indicates the significant growth inhibition of Wilac D001 with time mostly in acidic conditions so the growth curve with a higher time incubation is needed to show its long-term survival in pH 2 and pH 4.
→ In the reference we have cited [12], and other acidic tolerance experiments observed for 6 hours. Therefore, we have assumed 6 hours might be enough for testing the survival rate of W. confusa in different acidic conditions. Next time, if we have a chance to develop the test, we will check a longer time according to the growth curve.
Reference
- Guan, X.; Xu, Q.; Zheng, Y.; Qian, L.; Lin, B. Screening and characterization of lactic acid bacterial strains that produce fermented milk and reduce cholesterol levels. brazilian journal of microbiology 2017, 48, 730-739.
- While the study highlights the antifungal effects of Wilac D001 against VECs infected by C. albicans, discussing the broader implications of its efficacy in different clinical scenarios or against other microbial infections would enrich the discussion. Addressing the potential applicability of Wilac D001 beyond VVC could provide a more holistic perspective on its therapeutic potential. Overall, addressing these points would strengthen the paper and enhance its contribution to the field of antifungal therapy for VVC.
→ We have inserted a paragraph in page 9, line 295-304 –
“However, other W. confusa strains show promise in mitigating gut dysbiosis linked to inflammatory bowel disease and cancers, with demonstrated benefits in maintaining intestinal tight junctions and modulating intestinal cell responses. [13, 14, 15]. Therefore, the precise taxonomic category is crucial for understanding its potential pathogenicity. The strain what we used in this experiment is Wilac D001, which is a non-spore forming, gram-positive, coccobacillus that has obtained safety approval from the FDA. Wilac D001 has been annotated by the National Center for Biotechnology Information (NCBI, CP110106).”
Reference
- Jin, H.; Jeong, Y.; Yoo, S.-H.; Johnston, T.V.; Ku, S.; Ji, G.E. Isolation and characterization of high exopolysaccharide-producing Weissella confusa VP30 from young children’s feces. Microbial cell factories 2019, 18, 1-13.
- Ahmed, S.; Singh, S.; Singh, V.; Roberts, K.D.; Zaidi, A.; Rodriguez-Palacios, A. The Weissella genus: Clinically treatable bacteria with antimicrobial/probiotic effects on inflammation and cancer. Microorganisms 2022, 10, 2427.
- Fatmawati, N.N.D.; Gotoh, K.; Mayura, I.P.B.; Nocianitri, K.A.; Suwardana, G.N.R.; Komalasari, N.L.G.Y.; Ramona, Y.; Sakaguchi, M.; Matsushita, O.; Sujaya, I.N. Enhancement of intestinal epithelial barrier function by Weissella confusa F213 and Lactobacillus rhamnosus FBB81 probiotic candidates in an in vitro model of hydrogen peroxide-induced inflammatory bowel disease. BMC Research Notes 2020, 13, 1-7

Reviewer 3 Report
Comments and Suggestions for Authors
In their manuscript, the authors evaluated the effects of Wilac D001 on C. albicans using a vaginal epithelium model. Overall, the study was innovative, well-performed and critically conducted, using scientific-based methods. I have only a few minor points to be clarified before publication.
In general, the abstract is fine. Please, revise spelling, punctuations and structure such as in the following sentence: "Weissella confusa WIKIM51(Wilac D001) is known to be detected in dandelion kimchi produces lactic acid and anti-inflammatory ability, ...".
Also "P < 0.001, respectively". Respectively for what? There is only one value.
In the Introduction: "Recent in vitro and in vivo studies have demonstrated the lactobacilli mediated immune defense against C. albicans colonization [8,9]." References 8 and 9 are from 2017 and 1997, respectively. Thus, the mentioned studies are not recent. Please rephrase it.
Methods:
"Bacterial strains, including Wilac D001, L. reuteri, and L. rhamnosus, were supplied by Pharms Ville". Which strains did the authors use? Please clarify it.
The authors should provide details on the manufacturer for all the reagents and equipments used.
Disk diffusion test: The authors should give further details on how the inhibition zones have been measured.
Statistical analysis: The authors should mention if their data presented normality.
Conclusion: "Therefore, Wilac D001 can be considered as an effective group of lactobacilli for maintaining a healthy vaginal environment." The in vitro nature of the study does not support such a statement. Please rephrase it attenuating this sentence.
Comments on the Quality of English Language
Minimum revision required.
Author Response
Thank you very much for taking the time to review this manuscript-applsci-2844512. Please find the detailed responses below and the corresponding revisions and changes in the re-submitted files.
[Reviewer 3]
In their manuscript, the authors evaluated the effects of Wilac D001 on C. albicans using a vaginal epithelium model. Overall, the study was innovative, well-performed and critically conducted, using scientific-based methods. I have only a few minor points to be clarified before publication. In general, the abstract is fine. Please, revise spelling, punctuations and structure such as in the following sentence: "Weissella confusa WIKIM51(Wilac D001) is known to be detected in dandelion kimchi produces lactic acid and anti-inflammatory ability, ...".
→ I appreciated for your kind comments. I carefully corrected the abstract paragraph and general part.
Also "P < 0.001, respectively". Respectively for what? There is only one value.
→ I apologize that I had made a confusing. I meant that W. confusa was able to reduce either the concentrations of IL-6 and IL-8 independently and both of P values were under 0.001. I hope the exact meaning is conveyed. I have corrected that sentence for the clarity in page 1, line 26-28 –
“Both pro-inflammatory cytokines including IL-6 and IL-8 concentrations were significantly decreased when Wilac D001 was treated on C. albicans-infected VECs, respectively (P < 0.001).”
In the Introduction: "Recent in vitro and in vivo studies have demonstrated the lactobacilli mediated immune defense against C. albicans colonization [8,9]." References 8 and 9 are from 2017 and 1997, respectively. Thus, the mentioned studies are not recent. Please rephrase it.
→ I agree with this opinion and thank to your precise comments. I replaced them with recent in vitro (2020) [1] and in vivo (2024) paper [2].
Reference
- De Gregorio, P.R.; Parolin, C.; Abruzzo, A.; Luppi, B.; Protti, M.; Mercolini, L.; Silva, J.A.; Giordani, B.; Marangoni, A.; Nader-Macías, M.E.F. Biosurfactant from vaginal Lactobacillus crispatus BC1 as a promising agent to interfere with Candida adhesion. Microbial cell factories 2020, 19, 1-16.
- Bertarello, C.; Savio, D.; Morelli, L.; Bouzalov, S.; Davidova, D.; Bonetti, A. Efficacy and safety of Lactobacillus plantarum P 17630 strain soft vaginal capsule in vaginal candidiasis: a randomized non-inferiority clinical trial. European Review for Medical & Pharmacological Sciences 2024, 28.
Methods: "Bacterial strains, including Wilac D001, L. reuteri, and L. rhamnosus, were supplied by Pharms Ville". Which strains did the authors use? Please clarify it.
→ Thank you for your suggestion. I revised exact strain name in the method part page 2, line 88-90 –
“Bacterial strains, including Weissella confusa WIKIM51(Wilac D001), L. reuteri(RC-14), and L. rhamnosus (GR-1), were supplied by Pharms Ville (Seoul, Republic of Korea).”
Methods: "Bacterial strains, including Wilac D001, L. reuteri, and L. rhamnosus, were supplied by Pharms Ville". Which strains did the authors use? Please clarify it.
→ Thank you for your suggestion. I revised exact strain name in the method part page 2, line 88-90. “Bacterial strains, including Weissella confusa WIKIM51(Wilac D001), L. reuteri(RC-14), and L. rhamnosus (GR-1), were supplied by Pharms Ville (Seoul, Republic of Korea).”
The authors should provide details on the manufacturer for all the reagents and equipments used.
→ I have corrected method part with manufacturers information. These changes can be found in the revised manuscript overall material and method part- from page 2 to page 3 with red color.
Disk diffusion test: The authors should give further details on how the inhibition zones have been measured.
→ As you can see the below picture how to measure the inhibiting zones. According to this CDC standard method, we took a part of the pictures and measured the largest diameter for analysis.
Reference
Standard information from CDC Disk diffusion susceptibility:
https://www.cdc.gov/std/gonorrhea/lab/diskdiff.htm (assessed on 27th, Feb 2024)
Statistical analysis: The authors should mention if their data presented normality.
→ I confirmed that all the data were normalized with Shapiro-Wilk method. I inserted the phrase on page 4, line 172-173 –
“The Shapiro-Wilk test was performed to test normal distribution and all data satisfied normality.”
Conclusion: "Therefore, Wilac D001 can be considered as an effective group of lactobacilli for maintaining a healthy vaginal environment." The in vitro nature of the study does not support such a statement. Please rephrase it attenuating this sentence.
→ Thank you for pointing this out. I have corrected that conclusion considering your comments. Mentions can be found in page 10, line371-374, conclusion paragraph. –
“Therefore, Wilac D001 has an antifungal ability on VECs infected by C. albicans, especially by regulating pro-inflammatory cytokines. To investigate its efficacy as a health supplements further in vivo studies will be needed..”

Round 2
Reviewer 1 Report
Comments and Suggestions for Authors
This Reviewer acknowledges that in the re-submission, the authors have made several changes to attempt to improve the manuscript. However, in the final analysis, this remains a completely in vitro study with only a very general mechanism proposed for Wilac D001, which is biofilm disruption.
The growth of Wilac D001 is much higher at pH 7 (A600 ~ 0.3) than at pH 4 (A600 < 0.07), which is a problem because pH of the human vaginal tract is closer to the latter and with an observed A600 of <0.07 at 4 and 6 hr, it is unlikely that the Wilac D001 organism will achieve significant numbers in the human vagina for biofilm disruption. The situation at pH 7 is slightly better, although the A600 for Wilac D001 appears to peak at 0.3 at both 4 and 6 hr timepoints. In other words, Wilac D001 grows but des not thrive at pH 7. Still, perhaps the observed A600 value of 0.3 might correlate with 10^7 CFU/ml, as used in Figure 2, but that is not known or presented here.
It may be possible for an intravaginal infusion of a solution of 10^7 CFU/ml Wilac D001 to show antifungal efficacy for women, but an in vivo study in mouse or rat is first warranted, and we are far from that right now.
Other issues:
Lines 129-132: VECs were co-cultured with C. albicans (1x10^5 CFU/ml) and maintained for 36 h to induce inflammation [28].
It is not possible to induce inflammation in vitro. There is no immune system present in vitro.
Lines 104-105. Sigma is in MO, not MA, as far as I recall.
The adhesion test is carried out with 10^7 CFU/ml, but the cytokine analysis is carried out with 10^5 CFU/ml (see Methods, lines 116 and 126). Why are these two inoculation doses not harmonized?
Other Point of Interest
The entire argument that Wilac D001 disrupts Candida albicans biofilm formation is based on the adhesion assay with VECs in Figure 2 and the observation of decreased aggregation of Candida on the surface of the VECs. Is that sufficient to make the claim that Wilac D001 works by a mechanism that disrupts biofilm formation? Is decreased adhesion the single-most important aspect of disrupted biofilm formation? The black arrow in Panel 2E seems to be pointing to hyphal yeast. Did Wilac D001 have an effect on the number of hyphal yeast observed in your study?
Also, what happens if instead of 1x10^7 CFU/ml of Wilac D001, you use 1x10^6 CFU/ml or 1x10^5 CFU/ml in the adhesion study? Are the effects of Wilac D001 dependent on the inoculating dose? I ask this because if you look carefully at the images, you need about 10 Wilac D001 cells to equal one hyphal yeast cell. You seeded the same number of both Wilac D001 and C. albicans. Under that condition, C. albicans adhesion is completed suppressed.
The above comment really asks the authors how confident can they be that the mechanism of Wilac D001 involves disrupting biofilm formation? Have the authors explored the possibility that some kind of soluble mediator released by the Wilac D001 strain is disrupting the adhesion?
Comments on the Quality of English Language
Some of the revised text introduced here remains confusing in English.
lines 328-330
"As we can find in this result, antimicrobial can be the convincing strategy to reduce growth of C. albicans. However, Wilac D001 and lactobacillus spp. present significant antifungal effect compared to disk only."
Also, for lines 342-348 below: While the meaning comes through, the text should be re-written because it sounds strange. What does the author mean by survival rate? A rate is a variable over time. So, does the author mean A600/hr? I am confused.
"At pH2 media, Wilac D001 successively survived for 2 hours. Despite the survival rate was decreased from 4 hour, it still showed similar with 344 L. reuteri. In the pH4, the L. rhamnosus showed significantly high survival rate for 6 hours. Wilac D001 also showed significantly high survival rate at 4 hours. Although it was reduced for 6 h, still Wilac D001 can survive as much as L. reuteri, which indicate that Wilac D001 might survive in the acidic vaginal environment for 6 hours."
Author Response
Thank you very much for taking the time to review this manuscript-applsci-2844512. Please find the detailed responses below and the corresponding revisions and changes in the re-submitted files.
|
Point-by-point response to |
|
|
This Reviewer acknowledges that in the re-submission, the authors have made several changes to attempt to improve the manuscript. However, in the final analysis, this remains a completely in vitro study with only a very general mechanism proposed for Wilac D001, which is biofilm disruption.
The growth of Wilac D001 is much higher at pH 7 (A600 ~ 0.3) than at pH 4 (A600 < 0.07), which is a problem because pH of the human vaginal tract is closer to the latter and with an observed A600 of <0.07 at 4 and 6 hr, it is unlikely that the Wilac D001 organism will achieve significant numbers in the human vagina for biofilm disruption. The situation at pH 7 is slightly better, although the A600 for Wilac D001 appears to peak at 0.3 at both 4 and 6 hr timepoints. In other words, Wilac D001 grows but does not thrive at pH 7. Still, perhaps the observed A600 value of 0.3 might correlate with 10^7 CFU/ml, as used in Figure 2, but that is not known or presented here.
It may be possible for an intravaginal infusion of a solution of 10^7 CFU/ml Wilac D001 to show antifungal efficacy for women, but an in vivo study in mouse or rat is first warranted, and we are far from that right now.
→ I appreciate your valuable suggestion. It was a valuable time to review these experiments. We have decided to exclude the acid tolerance test after considering the issues you pointed out. The limitation is that the OD value alone may not sufficiently reflect the survival rate of viable bacteria, as the debris from dead bacteria can also be present in the media. Consequently, we have modified the manuscript to focus on presenting a comprehensive analysis of antifungal and anti-proinflammatory effects.
Other issues:
Lines 129-132: VECs were co-cultured with C. albicans (1x10^5 CFU/ml) and maintained for 36 h to induce inflammation [28].
It is not possible to induce inflammation in vitro. There is no immune system present in vitro.
→ We have corrected this sentence such as –
“VECs were co-cultured with C. albicans (1x10^5 CFU/ml) and maintained for 36 h to induce pro-inflammatory cytokine stimulation.”
Lines 104-105. Sigma is in MO, not MA, as far as I recall.
→ We have corrected.
The adhesion test is carried out with 10^7 CFU/ml, but the cytokine analysis is carried out with 10^5 CFU/ml (see Methods, lines 116 and 126). Why are these two inoculation doses not harmonized?
→ We apologize for confusing. We checked the experimental notes again, and the final CFU dose for cytokine assay is 10^7 CFU/100ul. We have corrected in manuscript.
Figure1. Experiment notes.
Other Point of Interest
The entire argument that Wilac D001 disrupts Candida albicans biofilm formation is based on the adhesion assay with VECs in Figure 2 and the observation of decreased aggregation of Candida on the surface of the VECs. Is that sufficient to make the claim that Wilac D001 works by a mechanism that disrupts biofilm formation? Is decreased adhesion the single-most important aspect of disrupted biofilm formation? The black arrow in Panel 2E seems to be pointing to hyphal yeast. Did Wilac D001 have an effect on the number of hyphal yeasts observed in your study?
Also, what happens if instead of 1x10^7 CFU/ml of Wilac D001, you use 1x10^6 CFU/ml or 1x10^5 CFU/ml in the adhesion study? Are the effects of Wilac D001 dependent on the inoculating dose? I ask this because if you look carefully at the images, you need about 10 Wilac D001 cells to equal one hyphal yeast cell. You seeded the same number of both Wilac D001 and C. albicans. Under that condition, C. albicans adhesion is completed suppressed.
The above comment really asks the authors how confident can they be that the mechanism of Wilac D001 involves disrupting biofilm formation? Have the authors explored the possibility that some kind of soluble mediator released by the Wilac D001 strain is disrupting the adhesion?
→ This result presents a reduction of colonization of C. albicans which was shown in Figure 1 Panel B. The gram-staining method was unable to elucidate the mechanism of biofilm disruption. Nevertheless, it did reveal the absence of any colonization by C. albicans so that these Phenomenological results can support the claim that Wilac D001 interrupt the C. albicans filamentation formation.
The progression of biofilm formation involves several stages: 1) adherence to a surface in the spherical yeast form, 2) the development of a basal layer comprising anchoring cells to initiate proliferation, 3) maturation characterized by the production of an extracellular matrix through the transformation from pseudohyphae to hyphal yeast form, and 4) eventual dispersal to seek new sites for colonization. Therefore, reduction by blocking the adherence of C. albicans to VECs and preventing its colonization are crucial initial steps in the biofilm formation process.
In Figure 2, only the filamentous pseudohyphae structure of C. albicans was observed without any signs of colonization when inoculated with Wilac D001 and Lactobacillus species. Additionally, Wilac D001 and Lactobacillus spp. were found to attach to the surface of C. albicans. Consequently, Wilac D001 exhibited an influence on the reduction of hyphal yeast numbers.
We agree with your comments that the inoculation dose may influence the effects of Wilac D001 and implications for the outcomes of this study. Further research, involving a comparative study with various inoculation doses and incubation times, is warranted to achieve consistent results. This is based in prior experiments that included an adhesion test spanning a range from 1x10^4 CFU/ml to 1x10^8 CFU/ml [1-4].
Also, previous study suggests some molecules secreted by Lactobacillus species can influence the attachment of bacteria to C. albicans and VECs [5-6]. Unfortunately, we have not investigated the components of the soluble mediator produced by Wilac D001. To explore the mechanism of the antifungal effect of Wilac D001 in future studies, ingredient analysis should be conducted.
Reference
- Osset, J.; Bartolome, R.M.; Garcia, E.; Andreu, A. Assessment of the capacity of Lactobacillus to inhibit the growth of uropathogens and block their adhesion to vaginal epithelial cells. J Infect Dis 2001, 183, 485-491, doi:10.1086/318070.
- Naidu, A.S.; Chen, J.; Martinez, C.; Tulpinski, J.; Pal, B.K.; Fowler, R.S. Activated lactoferrin's ability to inhibit Candida growth and block yeast adhesion to the vaginal epithelial monolayer. J Reprod Med 2004, 49, 859-866.
- Zárate, G.; Nader‐Macias, M. Influence of probiotic vaginal lactobacilli on in vitro adhesion of urogenital pathogens to vaginal epithelial cells. Letters in Applied Microbiology 2006, 43, 174-180.
- Lee, Y. Characterization of Weissella kimchii PL9023 as a potential probiotic for women. FEMS Microbiology Letters 2005, 250, 157-162.
- MacAlpine, J.; Daniel-Ivad, M.; Liu, Z.; Yano, J.; Revie, N.M.; Todd, R.T.; Stogios, P.J.; Sanchez, H.; O’Meara, T.R.; Tompkins, T.A.; et al. A small molecule produced by Lactobacillus species blocks Candida albicans filamentation by inhibiting a DYRK1-family kinase. Nature Communications 2021, 12, 6151, doi:10.1038/s41467-021-26390-w.
- Fatmawati, N.N.D.; Gotoh, K.; Mayura, I.P.B.; Nocianitri, K.A.; Suwardana, G.N.R.; Komalasari, N.L.G.Y.; Ramona, Y.; Sakaguchi, M.; Matsushita, O.; Sujaya, I.N. Enhancement of intestinal epithelial barrier function by Weissella confusa F213 and Lactobacillus rhamnosus FBB81 probiotic candidates in an in vitro model of hydrogen peroxide-induced inflammatory bowel disease. BMC Research Notes 2020, 13, 1-740. Khan, F.; Bamunuarachchi, N.I.; Pham, D.T.N.; Tabassum, N.; Khan, M.S.A.; Kim, Y.-M. Mixed biofilms of pathogenic Candida-bacteria: Regulation mechanisms and treatment strategies. Critical Reviews in Microbiology 2021, 47, 699-727.
Comments on the Quality of English Language
"As we can find in this result, antimicrobial can be the convincing strategy to reduce growth of C. albicans. However, Wilac D001 and lactobacillus spp. present significant antifungal effect compared to disk only."
Also, for lines 342-348 below: While the meaning comes through, the text should be re-written because it sounds strange. What does the author mean by survival rate? A rate is a variable over time. So, does the author mean A600/hr? I am confused.
"At pH2 media, Wilac D001 successively survived for 2 hours. Despite the survival rate was decreased from 4 hour, it still showed similar with 344 L. reuteri. In the pH4, the L. rhamnosus showed significantly high survival rate for 6 hours. Wilac D001 also showed significantly high survival rate at 4 hours. Although it was reduced for 6 h, still Wilac D001 can survive as much as L. reuteri, which indicate that Wilac D001 might survive in the acidic vaginal environment for 6 hours."
→ Considering your comprehensive suggestion, I have corrected these sentences in -
“As we can find this disk diffusion results, antimicrobials emerge as a compelling strategy for inhibiting the growth of C. albicans. Nevertheless, Wilac D001 and Lactobacillus spp. demonstrate a significant antifungal effect, surpassing that of the disk alone.”
The acidic tolerance paragraph has been deleted.

Reviewer 2 Report
Comments and Suggestions for Authors
The authors have provided satisfactory changes in the manuscript, Now it can be accepted in its revised form.
Author Response
The authors have provided satisfactory changes in the manuscript, Now it can be accepted in its revised form.
>> Thank you very much for taking the time to review this manuscript-applsci-2844512. It was valuable time to reveiw this manusscript
Round 3
Reviewer 1 Report
Comments and Suggestions for Authors
Again, this is a very introductory study that has limited value but can become very intriguing with more development.
Comments on the Quality of English LanguageMost of the English issues have been addressed, but read through it once more to be sure.